



# Non-reversible aging can increase solar absorption in African biomass burning aerosol plumes of intermediate age

Amie Dobracki[1], Paquita Zuidema[1], Steve Howell[2], Pablo Saide[3], Steffen Freitag[4], Allison C. Aiken[5], Sharon P. Burton[6], Arthur J. Sedlacek III[7], Jens Redemann[8], and Robert Wood[9]

[1]University of Miami, Miami, Florida, USA
[2]University of Hawai'i at Mānoa, Honolulu, USA
[3]University of California Los Angeles, Los Angeles, California, USA
[4]University of Hawai'i at Mānoa, Honolulu, USA
[5]Earth and Environmental Sciences Division, Los Alamos National Laboratory, Los Alamos, New Mexico, USA
[6]NASA Langley Research Center, Hampton, VA, USA
[7]Brookhaven National Laboratory, Upton, New York, USA
[8]University of Oklahoma, Norman, Oklahoma, USA
[9]University of Washington, Seattle, WA, USA

**Correspondence:** Paquita Zuidema (pzuidema@miami.edu) and Amie Dobracki (amie.dobracki@rsmas.miami.edu)

**Abstract.** Recent studies highlight that biomass-burning aerosol over the remote southeast Atlantic is some of the most sunlight-absorbing aerosol on the planet. *In-situ* measurements of single-scattering albedo at the 530 nm wavelength ($SSA_{530nm}$) range from 0.83 to 0.89 within six flights (five in September, 2016 and one in late August, 2017) of the ORACLES (ObseRvations of Aerosols above CLouds and their intEractionS) aircraft campaign, increasing with the organic aerosol to black carbon (OA:BC) mass ratio. OA:BC mass ratios of 10 to 14 are lower than some model values and consistent with BC-enriched source emissions, based on indirect inferences of fuel type (savannah grasslands) and dry, flame-efficient combustion conditions. These primarily explain the low single-scattering albedos. We investigate whether continued chemical aging of aerosol plumes of intermediate age (4-7 days after emission, as determined from model tracers) within the free troposphere can further lower the $SSA_{530nm}$. A mean OA to organic carbon mass ratio of 2.2 indicates highly oxygenated aerosol with the chemical marker *f44* indicating the free-tropospheric aerosol continues to oxidize after advecting offshore of continental Africa. Two flights, for which BC to carbon monoxide (CO) ratios remain constant with increasing chemical age, are analyzed further. In both flights, the OA:BC mass ratio decreases over the same time span, indicating continuing net aerosol loss. One flight sampled younger (∼ 4 days) aerosol within the strong zonal outflow of the 4-6 km altitude African Easterly Jet-South. This possessed the highest OA:BC mass ratio of the 2016 campaign and overlaid slightly older aerosol with proportionately less OA, although the age difference of one day is not enough to attribute to a large-scale recirculation and subsidence pattern. The other flight sampled aerosol constrained closer to the coast by a mid-latitude disturbance and found older aerosol aloft overlying younger aerosol. Its vertical increase in OA:BC and nitrate to BC was less pronounced than when younger aerosol overlaid older aerosol, consistent with compensation between a net aerosol loss through aging and a thermodynamical partitioning. Organic nitrate provided 68% on average of the total nitrate for the 6 flights, in contrast to measurements made at Ascension Island that only found inorganic nitrate. Some evidence for the thermodynamical partitioning to the particle phase at higher altitudes with higher





relative humidities for nitrate is still found. The 470-660 nm absorption Angstrom exponent is slightly higher near the African coast than further offshore (approximately 1.2 versus 1.0-1.1), indicating some brown carbon may be present near the coast. The data support the following parameterization: $SSA_{530nm}$=0.80+0056*(OA:BC). This indicates a 20% decrease in SSA can be attributed to chemical aging, or the net 25% reduction in OA:BC documented for constant BC:CO ratios.

# 1  Introduction

Biomass burning, the largest source of carbon to the atmosphere globally, is fundamental to the Earth's global carbon cycle (Bowman et al., 2009; Bond et al., 2013). The emitted carbonaceous aerosols and trace gases include the greenhouse gases CO and $CO_2$, and significantly alter the atmospheric composition over large regions of the globe (Andreae, 2019). This in
turn influences all of the gaseous, aerosol and aerosol-cloud interaction radiative forcing terms considered by the latest IPCC Assessment. Despite the importance of biomass burning events on climate, how the properties of smoke change following long-range transport are still largely unknown. Smoke emissions undergoing long-range transport include the effluents from northern European and Russian forest fires reaching the Arctic basin (Cubison et al., 2011), wildfire smoke from western continental north America observed over Europe (Zheng et al., 2020; Baars et al., 2021), and aerosols from fires in southern
Africa reaching south America (Holanda et al., 2020). Without wet or dry scavenging, the aerosol's areal coverage is increased through transport. As the carbonaceous aerosol-dominated plume is advected, it undergoes chemical, optical and physical changes. Black carbon (BC) is largely inert but is accompanied by primary organic aerosol at emission that varies with fuel type and combustion conditions (flaming versus smouldering). The subsequent formation of secondary organic aerosol (SOA) provides additional mass in under two days through a condensation of gases, aqueous-phase chemistry (if sufficient moisture
is available) and photochemistry (Jimenez et al., 2009). Further oxidation chemistry and photolysis can provide a subsequent SOA sink (Wagstrom and Pandis, 2009; Cubison et al., 2011; Collier et al., 2016; Ahern et al., 2018; Cappa et al., 2020), prior to the aerosol's ultimate removal by deposition. As such, many processes can influence the OA:BC ratios. Many global aerosol models overestimate OA concentrations in the remote free troposphere (Heald et al., 2010, 2011; Spracklen et al., 2011; Tsigaridis et al., 2014; Hodzic et al., 2015, 2016; Shinozuka et al., 2020). Models with more realistic wet removal rates
can compare better to observations within convective environments (Cape et al., 2012; Katich et al., 2018; Lund et al., 2018; Hodzic et al., 2020), but this neglects loss processes occurring in non-convective regions.

Southern Africa produces approximately one-third of the world's fire-emitted carbon (van der Werf et al., 2010). The global maximum of absorbing aerosol above cloud occurs above the southeast Atlantic (Waquet et al., 2013), so that the aerosols provide a distinct radiative warming over the ocean (Keil and Haywood, 2003; Graaf et al., 2014; Zuidema et al., 2016b;
Mallet et al., 2021; Doherty et al., 2021). BBA from this region is unusual for being highly absorbing of sunlight, with recent studies reporting SSA values of 0.85 or less at the green wavelength (Zuidema et al., 2018; Chylek et al., 2019; Pistone et al.,





2019; Holanda et al., 2020; Taylor et al., 2020; Denjean et al., 2020; Mallet et al., 2020; Shinozuka et al., 2020; Carter et al., 2021), lower than what is currently representative of many global and regional models (Shinozuka et al., 2020; Mallet et al., 2021). This has significant implications for regional climate and its circulation (e.g., Mallet et al., 2020; Solmon et al., 2021).

While climate models discern an ensemble-mean direct radiative warming, individual models disagree strongly on magnitude and even sign (Myhre et al., 2013; Zuidema et al., 2016b; Haywood et al., 2021; Mallet et al., 2021). In addition, the direct aerosol radiative effect estimated from satellites typically exceeds model estimates (de Graaf et al., 2020). That the measured single-scattering albedos are lower than what is currently implemented in many models (Mallet et al., 2021), suggests one cause for an underestimated modeled direct radiative warming.

This study's primary explanation for why the BBA is so highly absorbing of sunlight is that the aerosols are already highly absorbing due to a high black carbon content at emission. A survey of the published emission factors for the vegetation types typical of southern Africa - savannahs, grasslands, agricultural fields, and at times tropical forest (Table 1) indicates that the high black carbon to carbon monoxide (BC:CO) ratios reported later in this study are primarily representative of savannah burning. Savanna fires typically produce more black carbon than do agricultural or woodland fires. (Andreae, 2019). Daily maps of fire

locations for the flight days (not shown) suggest that the BBA sources are primarily fire emissions from miombo woodlands, which contain a significant fraction of savanna grasses and some agricultural fields (Shea et al., 1996; Christian et al., 2003; Korontzi et al., 2003; Vakkari et al., 2018; Huntley, 2019), distributed over a broad geographic region encompassing Angola, Tanzania and the Congo (Carter et al., 2021; Redemann et al., 2021).

Despite the efficiency at which grasslands can burn, southern African fires can also produce significant near-source secondary

organic aerosol (SOA), depending on the burning conditions (Vakkari et al., 2018; Pokhrel et al., 2021). We further investigate whether continued chemical aging of BBA after it has advected away from the African continent may further lower the aerosol SSA, primarily by reducing the OA content. Because the aerosol now overlies the brighter surface of the stratocumulus deck, any additional increase in solar absorption can significantly increase the regional climate warming by the aerosol (e.g., Keil and Haywood, 2003). In this sense this study complements that of Wu et al. (2020), who examined more fully aged aerosol

at Ascension Island (8°S, 14.5°W) and concluded that variations in the single-scattering albedo with altitude were primarily related to the thermodynamical repartitioning of inorganic nitrate. This explains why the aerosol sampled at higher altitudes at Ascension have a higher SSA, because the inorganic nitrate will favor the particle phase at the higher altitude. More importantly, this indicates a reversible process, in contrast to the irreversible reduction in the SSA from a loss of organic aerosol (including organic nitrate) we will try to show here.

We draw on *in-situ* smoke aerosol composition and optical property from six flights of the NASA Earth Venture Suborbital-2 ORACLES (ObseRvations of Aerosols above CLouds and their intEractionS) (Redemann et al., 2021) aircraft deployments, shown in Fig. 1. The selected flights occur within 30 days of each other within the seasonal cycle, to improve the likelihood that the composition of the fire emissions remained similar. Black carbon and carbon monoxide, which are both conserved after emission, are used to infer homogeneity in the fire sources. In one flight, a region of enhanced zonal winds occurring between 4-

6 km at approximately 10°S, known as the African Easterly Jet-South (Adebiyi and Zuidema, 2016; Pistone et al., 2021; Ryoo et al., 2021), advected an aerosol layer with the highest OA:BC mass ratio measured during the 2016 campaign (approximately





14) offshore, where it overlaid an aerosol layer of a lower OA:BC mass ratio (Fig. 2). The upper OA-enriched aerosol is younger than the aerosol sampled at a lower altitude, based on an aerosol age (time since emission) estimated from tracers released within the campaign's aerosol forecast model (shown later). This could be consistent with an anticyclonic circulation

recirculating the fresher aerosol aloft while slowly subsiding back to the African continent (see Fig. 21 of Redemann et al., 2021), losing organic aerosol along the way. We note a small net OA loss has previously been reported from other African BBA aircraft campaigns (Jolleys et al., 2012). If representative of a larger-scale aerosol-meteorology environment, this would help explain an increase in SSA with altitude anecdotally noted in the field that is separate from that caused by the thermodynamical phase partitioning of inorganic nitrate (Wu et al., 2020).

**2   Datasets**

### 2.1   Weather Research and Forecasting Aerosol Aware Microphysics (WRF-AAM) Model

Model-derived estimates for the mean aerosol time since emission (age) of up to two weeks were calculated using the campaign's operational aerosol forecast model, the Weather Research and Aerosol Aware Microphysics (WRF-AAM) Model (Thompson and Eidhammer, 2014). The model releases tracers tagged to carbon monoxide (CO) at the fire source for each

day of a two-week forecast. The fire source is taken from a burned area product of 500 m spatial resolution from the Moderate Resolution Imaging Spectrometer (Giglio et al., 2006) and may miss smaller burned areas. The current analysis takes advantage of the model's prior use for seeking out the smoke layers that are subsequently sampled by the aircraft. The model configuration is similar to that in Saide et al. (2016), with the 12-km spatial resolution regional model encompassing a domain (41°S-14°N, 34°W-51°E) sufficiently large to capture almost all contributing fires. The model is driven by the National Center

for Environmental Prediction Global Forecasting System meteorology, using daily smoke emissions from the Quick Fire Emissions Dataset (Darmenov and da Silva, 2013) released into the model surface layer. These are advected thereafter according to the model physics, with their spatial distribution constrained near real-time with satellite-derived optical depths. This allows a diurnal cycle representation of the daytime burning. The capabilities of WRF-AAM include photochemistry, particulate matter, optical properties, radiative forcing, aerosol radiation and cloud chemistry feedback, and aerosol-cloud interactions (Grell

et al., 2005; Wang et al., 2015).

### 2.2   Modified combustion efficiency

CO and $CO_2$ are used to infer fire emission conditions (Collier et al., 2016; Yokelson et al., 1997) through the modified combustion efficiency (MCE) metric:

$$MCE = \frac{\Delta CO_2}{\Delta CO + \Delta CO_2} = \frac{1}{1 + \Delta CO/\Delta CO_2} \tag{1}$$

A regression is used to estimate the $\Delta CO/\Delta CO_2$ with $\Delta CO$ and $\Delta CO_2$ representing the measured CO and $CO_2$ amounts, in moles, relative to background values. Higher values of MCE (>0.9) are associated with flaming combustion, whereas values less than 0.9 are more typical of smoldering combustion, for which more particles are typically emitted for the same amount of





fuel. Adopted background values were 65 (77) ppbv for CO, and 397 (404) ppmv for $CO_2$, in September 2016 (August 2017), based on measurements in the free troposphere taken above the smoke plumes ( 7000m).

Carbon monoxide was measured with an aircraft modified gas-phase $CO/CO_2/H_2O$ Analyzer from Los Gatos Research, operated and analyzed by NASA Ames (Jim Podolske). The analyzer uses a patented Integrated Cavity Output Spectroscopy (ICOS) technology to make stable cavity-enhanced absorption measurements of CO, $CO_2$, and $H_2O$ in the infrared spectral region. The instrument reports CO mixing ratio (mole fraction) at a 1-Hz rate based on measured absorption, gas temperature, and pressure using Beer's Law (Zellweger et al., 2012). The measurement precision is 0.5 ppbv over 10 seconds.

## 2.3   Aerosol Composition

The Hawaii Group for Environmental Aerosol Research (HiGEAR) operated an Aerodyne High-Resolution Time-of-Flight Aerosol Mass Spectrometer (HR-ToF-AMS, referred to as AMS) during ORACLES, building on previous experience in the southeast Pacific (Yang et al., 2011; Shank et al., 2012) and the Arctic (Howell et al., 2014). The native time resolution is approximately five seconds, with the data interpolated onto a one-second temporal grid to facilitate integration with other
datasets. The overall uncertainty in the reported aerosol mass concentrations is estimated at 33% to 37%, at a one-minute time resolution, based on Bahreini et al. (2009). Some analyses were restricted to level legs ranging from 4 to 10 minutes, listed in Table S1, further reducing the uncertainty about the mean to 19%-10 %. Further details on the AMS data treatment and the sampling layout are provided in the Supplement.

In addition the uncertainty in the OA:BC mass ratios are expected to be smaller when aerosol concentrations are larger,
because of improved signal-to-noise but also towards minimizing effects from dilution, by which OA evaporates through mixing with cleaner environmental air (e.g., Hodshire et al., 2021), and from model-observational disparities in the smoke plume locations, which are likely to be more noticeable at the smoke plume edges. We focus on those OA:BC mass ratios that have become stable with increasing OA mass (Fig. 5). The OA:BC mass ratio is significantly less for air with OA>3 $\mu$g $m^{-3}$ than for air with OA>20 $\mu$g $m^{-3}$, particularly for younger aerosol (Fig. 3a). Fig. 3b indicates that the OA:BC mass ratio
stabilizes at OA mass concentrations $\geq$ 20 $\mu$g $m^{-3}$, establishing the threshold we apply throughout this study. We note that Fig. 5a also indicates OA:BC mass ratios can increase again 10 days after emission, but do not pursue this as the model skill in the smoke plume locations is likely to become less over time.

Measurements of *f44*, the fraction of the OA mass spectrum signal at *m/z 44* relative to the total OA mass concentration, indicates the presence of the $CO_2^+$ ion, or oxidation resulting from chemical aging (Canagaratna et al., 2015). *f44* serves as a
robust physically-based proxy for BBA age and forms one of two metric for continuing aging. Elemental analysis, yielding hydrogen (H), oxygen (O) and organic carbon (OC) rely on the algorithms developed within Aiken et al. (2007).

Measurements of the black carbon mass concentration derive from a 4-channel single particle soot photometer (SP2, Droplet Measurement Technology) deployed by HiGEAR in 2016, and an 8-channel SP2, which allows further resolution of the black carbon coatings, deployed by Art Sedlacek of Brookhaven National Laboratory for the August 31 2017 campaign, with the
data processed by the two groups respectively. The SP2 uses laser (1064nm) induced incandescence for quantitative detection of refractory black carbon particles of mass equivalent diameter between approximately 80-500 nm in real time. The SP2 was





calibrated using fullerene soot effective density estimates from Gysel et al. (2011). Further details can be found in Schwarz et al. (2006).

Ratios of $\frac{\Delta BC}{\Delta CO}$ (shortened to BC:CO) serve to assess homogeneity of the aerosol composition at the source emission. The ratios are non-dimensionalized by using the ideal gas law at standard temperature (273K) and pressure (1000 hPa) to convert the CO concentrations from ppb to ng m$^{-3}$.

### 2.4 Determination of organic/inorganic nitrate contribution

Farmer et al. (2011) provide an approach for estimating the contribution to the total nitrate signal from organic nitrate (ON) using the $NO^+ : NO_2^+$ ratio, building on the observation that organic nitrates typically fragment into larger proportions of $NO^+$ than do inorganic nitrates (in their study, organic $NO^+$ ratios vary between 1.8 to 4.6 for different organonitrates, compared to 1.5 for $NH_4NO_3$). Their Equation 1, reproduced below, provides an estimate of the ON fraction that can be readily applied to the ORACLES AMS data, assuming that the ON fraction can be resolved. The success of this approach also assumes that the inorganic nitrates capable of providing a large $NO^+$ ratio, such as mineral nitrates, are not present. Both assumptions are justified for the SEA free troposphere.

$$X(ON\%) = \frac{(R_{obs} - R_{NH_4NO_3})(1 + R_{ON})}{(R_{ON} - R_{NH_4NO_3})(1 + R_{obs})} \tag{2}$$

$R_{obs}$ is the ORACLES *m/z* ratio of ion fragments 30 to 46, $R_{NH_4NO_3}$ is the ionization efficiency (IE) calibration-derived ratio (1.26 for 2016 and 1.545 for 2017) and an $R_{ON}$ value of 3.41 is a reference ratio based on the average fragmentation pattern into the $NO^+ : NO_2^+$ ratios for the OIA-HN, OIA-CN and OIA-olig standards evaluated within Table S1 of Farmer et al. (2011). The inorganic nitrate (IN) fraction is 1-ON. We use this approach to estimate the IN (primarily $NH_4NO_3$) fraction, keeping in mind that it is an indirect inference.

### 2.5 Optical Instruments: Nephelometers and Particle Soot Absorption Photometers

This study primarily focuses on the SSA at 530 nm wavelength, but also examines the absorption Angstrom exponent (AAE) and mass absorption cross-section (MAC) measurements for evidence of brown carbon. Scattering from all particles is measured continuously by a TSI nephelometer (model 3563) at the (470, 550, 700) nm wavelengths ($\lambda$), with a linear regression in log-log space used to estimate the scattering at 530 nm wavelength. The spectral light absorption coefficients ($\sigma_a$) of the total aerosol are measured by a Particle Soot Absorption Photometer (PSAP; Radiance Research) at the 470, 530, and 660 nm wavelengths. The values are an average based on measurements from two PSAPs in 2016, and only one PSAP functioned in 2017. Both filter-based measurements are corrected according to Anderson and Ogren (1998). The SSA values are based on the wavelength-averaged (as opposed to wavelength-length-specific) corrections of Virkkula (2010). The use of the average wavelength-corrected values reduces a potential high bias at the shortest wavelength introduced by multiple scattering within the PSAP signal (Pistone et al., 2019), and reduces spurious effects from filter changes (Zuidema et al., 2018). Although similar to Pistone et al. (2019), a stricter aerosol threshold (OA>20 $\mu$g m$^{-3}$ rather than scattering at 530nm > 10 Mm$^{-1}$) is applied and no arithmetic weighting by extinction is done. SSA values at 530 nm are at standard temperature and pressure.





The nephelometer measurements occurred at 40-50% relative humidity, while the PSAP measurements measured at a lower

$\sim$ 20% RH, brought about by heating the PSAP optical block to approximately 50°C (Pistone et al., 2019). Ambient RH measurements ranged up to 80%, with higher RH data samples excluded by construction. Two other single-wavelength (550 nm) nephelometers (Radiance Research, M903) measured at two different relative humidities, one at 80% and the other at below 40% RH (Howell et al., 2006). The impact on light scattering, estimated from the ratio of the ambient to dry RH measurements, is estimated to be less than 1.2 for 90% of the time within Shinozuka et al. (2020). The 20% increase in

scattering by the ambient RH is an upper bound, as the ambient RH was typically <80%. Aerosol absorption can also increase because of humidification (see discussion in Pistone et al. (2019)), introducing a compensating effect on the SSA, but this is likely smaller.

The absorption Angstrom exponent (AAE) values are calculated from the linear fit of $\log(\sigma_a)$ to $\log(\lambda)$. The mass absorption cross-section (MAC) measurements are based on the absorption coefficient divided by the BC mass concentration at 660 nm,

and by the BC+OA mass concentration at 470 nm. At 660 nm, the solar absorption is held to be dominated by the black carbon. Should brown carbon contribute to the solar absorption, this should affect the $\text{MAC}_{BC+OA}$.

## 3 Approach

### 3.1 Determination of Aging

Two independent measures of aerosol age with differing merits are invoked. Model tracers of CO released at the emission

source provide a precise but potentially uncertain time estimate while the oxidation information provided by the chemical marker *f44* is a clearer indicator of chemical aging. Model-derived trajectories become more uncertain over longer time spans and distances as the small differences in the model meteorology from nature accumulate. In addition, mixing at the aerosol plume edges can appear as an older aerosol age. Importantly, the measurements of *f44* correlate well with the younger model-derived age estimates (shown later), supporting the use of both metrics as aerosol age indicators.

### 3.2 Flight Selection

Flight selection is based on the availability of at least 20 minutes of organic aerosol (OA) data exceeding >20 $\mu$g m$^{-3}$ within the free troposphere (altitudes above 1.5 km) at relative humidities (RH) < 80%, and must possess aerosol model ages between 4-10 days for each flight. Towards improving the likelihood of sampling from similar aerosol source regions, only flights spanning August 31 through all of September for 2016 (5 flights) and 2017 (one flight) are considered, shown relative to the

satellite-derived above-cloud aerosol optical depths for September 2016 in Fig. 2. Aerosol forecast maps indicate the spatial sampling of the aerosol plumes for each flight (Figs. 4-5), with the OA data and model-estimated age displayed on individual altitude-latitude flight track projections. OA mass concentrations are often highly stable on individual level legs. The model-derived days since emission are almost always greater than four, suggesting much of the initial condensational growth of SOA from gases has ended by the time the BBA was sampled by the research aircraft.





The aircraft either flew along a routine southeast to northwest track (31 August, 4 and 25 September of 2016), or, performed target-of-opportunity flights sampling more aerosol-rich locations (6 and 24 September of 2016, 31 August 2017). The flight tracks make clear that the aircraft sampled widely, but never near the fire emission source. The aerosol spatial distribution was typically either constrained to near the coast, or elongated zonally along $10°$S (Figs. 4-5). During three of the flights (24-25 September 2016 and 31 August 2017), zonal easterly winds exceeded 6 m s$^{-1}$ along $\sim 10°$S at altitudes between 3-5 km. A

wind isotach, known as the African Easterly Jet-South (Nicholson and Grist, 2003; Adebiyi and Zuidema, 2016), explains the more zonally-elongated, meridionally-constrained aerosol plume spatial structure extending over the ocean evident in Fig. 5. Some recirculation back to land around the south Atlantic anticyclone also apparent for these three days (see also Ryoo et al. (2021) for more synoptic detail). Days in which the aerosol is constrained closer to the coast (Fig. 4) occur when a mid-latitude disturbance passes by to the south, altering the circulation to its north (Diamond et al., 2018; Kuete et al., 2020; Zhang and

Zuidema, 2021).

    The different flights intersect air of different ages, but none with model-estimated ages of less than 4 days. This is older aerosol than the oldest aerosol sampled during the Southern African Regional Science Initiative (SAFARI) campaign (Haywood et al., 2003), estimated at 2-3 days of age, but is younger than that sampled by the CLoud-Aerosol-Radiation Interaction and Forcing: Year-2017 (CLARIFY-2017 Haywood et al., 2021) over Ascension in August-September 2017 (Wu et al., 2020;

Taylor et al., 2020). The most data for aerosol aged 4-5 days since emission comes from just one flight, held on 24 September, 2016, highlighted in Fig 2. This flight's objective was to sample fresher aerosol within the AEJ-S as closely as possible to a fire emission source. Significant vertical gradients in aerosol age are evident in its four vertical profiles, with younger aerosol residing over older aerosol. The 31 August 2017 flight is included, despite being from another year, because the flight falls within the day-of-the-year selection from 2016, and because the flight sampled an aerosol layer of significant mass and

extremely stable OA:BC composition. The interannual variation in aerosol loading over the SEA is primarily driven by when the climatological AEJ-S sets in, as this is responsible for its geographic distribution of aerosol at higher altitudes. The AEJ-S, though weaker than climatology in August 2017 (Zhang and Zuidema, 2021; Ryoo et al., 2021), was present on 31 August, 2017.

## 4    Chemical composition and age distribution within the six flights

Mean *f44* values exceed 0.175 after 4 days of model-estimated age since emission (Fig. 6). These values are on par with *f44* values from Asian/Siberian smoke transported to Alaska over two weeks (Cubison et al., 2011) and indicate highly-oxidized aerosol. *f60* values are relatively constant and below 0.005. Both are consistent with chamber studies reporting lifetimes of *f44* and *f60* of approximately 20 days and 10 hours, respectively (George and Abbatt, 2010; Hodshire et al., 2019). Some continuing oxidation is indicated between model-derived days since of emission of 4-5 to days 5-6 (mean *f44* values of 0.175

increasing to 0.21), but no further change in *f44* is evident after 6 days since emission.

    Closely related to the average carbon oxidation state, flight-mean O:C mass ratios (Kroll et al., 2011) range between 0.61 to 0.69 for the 2016 flights, with small within-flight standard deviations (0.03-0.06) (Fig. 7). These values indicate a mixture of





(aged) low- and semi-volatile oxygenated OA (Jimenez et al., 2009; Huffman et al., 2009; Hodzic et al., 2020). The slope of the H:C to O:C mass ratios (Fig. 7; based on vanKrevelen (1950) describes the evolution of oxygenated organic aerosol. Downward movement along the slope of -1 is consistent with continuing oxidation through the formation of carboxylic acid ($C_6H_5OH$) formation, a process that ultimately replaces one hydrogen atom with one oxygen atom (Heald et al., 2010). Different SOA precursors from different sources or burning conditions may contribute to the range of the observed H:C ratios (Jimenez et al., 2009; Ng et al., 2011). Data from the 31 August, 2017 flight suggests more highly oxidized aerosol than any of the 2016 flights, with a mean O:C ratio of 0.81. The higher O:C mass ratio compared to the values from 2016 is not explained through the change in calibration constants between the two years, and related (we presume) to different fuel sources and conditions. Overall, the average (± standard deviation) plume values of H:C, O:C, and the organic-aerosol-to-organic-carbon mass ratio (OA:OC) are 1.2 ±0.1, 0.7 ±0.1, and 2.2±0.1, respectively, over all six flights. The OA:OC mass ratio, a measure of the oxygen content that is useful for model evaluation (Hodzic et al., 2020; Lou et al., 2020), is higher than common model-applied values of 1.4-1.8 for SOA:OC (Aiken et al., 2008; Tsigaridis et al., 2014; Hodzic et al., 2020) and primary OA:OC ratios measured near-source of 1.6 (Andreae, 2019), but are on par with measurements from the Atmospheric Tomography (ATom) campaign made in the same region (Hodzic et al., 2020).

Fig. 8 provides an overview of the range of values of *f44*, modified combustion efficiency (MCE), model-derived time since emission (age) and the non-dimensionalized BC:CO ratios for each of the flights. In combination, these indicate aged, oxidized aerosol emanating from flame-efficient fires with relatively high black carbon and low carbon monoxide efficiencies, without any strong outliers amongst the flights. *f44* flight-mean values range from 0.18 to 0.22. Modified combustion efficiency values are above 0.97 for each flight, indicating flame-efficient fires (Collier et al., 2016; Zhou et al., 2017) that are typical for grasslands and savannahs (Janhäll et al., 2010; Vakkari et al., 2018). Mean non-dimensionalized BC:CO ratios vary between 0.008 to 0.011, with a minimum on 24 September. The model-derived age estimates indicate most of the aerosol was emitted at least six days previously, with that on 9/24/2016 being the youngest, corresponding to the lowest *f44* values. The 8/31/2017 flight, for which Fig. 7 suggests the highest oxidation, has one of the higher mean *f44* values, but also one of the highest mean MCE and a higher mean BC:CO value, suggesting potential fuel type differences as well. OVerall, BC:CO ratios do not increase with increasing MCE as expected based on Kondo et al. (2011), but this likely reflects this study's small range of MCE values, for which Vakkari et al. (2018) also do not find a correlation. The mean values hint at a decrease in BC:CO over the 2016 BB season, consistent with the speculation within Eck et al. (2013) that the more combustible fuel may be ignited earlier in the BB season, but the trend is statistically-insignificant. The BC:CO ratios from the ORACLES and CLARIFY campaigns (Wu et al., 2020) are the highest of those shown for the surveyed literature in Table 1, exceeding those reported for southern Africa in Andreae (2019), and (Formenti et al., 2003). We note that Andreae (2019) indicate a higher emission factor for black carbon from savannah fires than Andreae and Merlet (2001), and, that the Akagi et al. (2011) emission factors, incorporated into many of the CMIP6 models (AR6, WG1, sec 6.2.2.6), produce higher BC:CO values for crop residue than for savannahs, in contrast to Andreae (2019).

MCE varies inversely with the moisture content for grasses (Korontzi et al., 2003), with leaf litter and woody fuels tending to dry more slowly than do grasses, and to burn by smoldering, as opposed to flaming. Thus we interpret the high MCE values



to reflect a large contribution from dry and dead grasses, as opposed to green grass and woodlands. The high BC:CO values for ORACLES and CLARIFY are also consistent with the burning of dry grass, which produces relatively low emissions of carbon

monoxide (Scholes et al., 1996). That the BC:CO values measured at offshore locations exceeds those measured previously over land suggests a possible coincidence between when the burning of the savannahs occurs with when the free-tropospheric winds capable of transporting the aerosols far to the west are more pronounced. Differences between aerosols in the free troposphere versus the boundary layer may also not be fully accounted for. Overall the mean submicron mass fractions of the six flights combined are 66% OA, 10% nitrate ($NO_3$), 11% sulfate ($SO_4$), 5% ammonium ($NH_4$), and 8% BC. $SO_4$:BC remains

constant at approximately 1.6-1.7 (not shown), indicating its formation as a secondary inorganic aerosol has ended.

## 5  Evidence for loss of organic aerosol with further oxidation

Given that secondary aerosol formation is expected to proceed more quickly when BC:CO ratios are lower (Vakkari et al., 2018), because the precursor gases may be more available (Yokelson et al., 2009), it is important to control for the BC:CO ratio if wishing to attribute OA:BC changes to aging rather than source differences. Fig. 9 shows the BC:CO ratios for each flight as

a function of *f44*. As expected, BC:CO can either increase or decrease with *f44*. Two flights show statistically-similar BC:CO values over a range of *f44* values, which we interpret to mean that source emissions remain similar over multiple days. These are the 8/31/2016 flight, for which BC:CO values range from 9.8-10.0 ($*10^{-3}$) for *f44* values of 0.18-0.27, and the 9/24/2016 flight, for which BC:CO values range from 7.5-7.9 ($*10^{-3}$) for *f44* values of 0.15-0.24.

The OA:BC mass ratios are broken down by *f44* for each flight in Fig. 10. The OA:BC mass ratios vary more significantly

between the flights than do the BC:CO ratios. Decreases in OA:BC are evident for both data segments with near-constant BC:CO identified in Fig. 9. For the 8/31/2016 flight, the OA:BC ratio decreases from 9.3 to 7.4 - a decrease of approximately 25% over the span of 2-3 days (based on Fig. 8), whereas for the 9/24/2016 flight, the OA:BC ratio decreases from 14.2 to 9.8 as *f44* values increase from 0.15-0.18 to 0.21-0.24 - an approximate 35% decline over a span of 2-3 days. Given that the BC:CO values did not change as a function of *f44*, we interpret the decrease in OA:BC as the aerosol continues to oxidize to mean that

a net OA loss continues after the aerosol has left the African continent. The higher OA:BC mass ratios on 24 September are consistent with lower the BC:CO values for this day, compared to 31 August 2016.

## 6  Thermodynamical repartitioning of aerosol versus chemical loss

An additional consideration could be that if the older aerosol is also situated lower within the atmosphere, that a smaller OA:BC mass ratio at a lower altitude could in fact reflect a thermodynamical particle-to-gas repartitioning, as opposed to an irreversible

loss of OA. This is investigated further for the same two flights (8/31/2016 and 9/24/2016). The selected aircraft profiles sampled air with different vertical structures in aerosol age, which we contrast in search of insights. The thermodynamic behavior would be most apparent in the inorganic nitrate fraction (e.g., Wu et al., 2020) and we examine the inorganic nitrate partitioning within the two individual profiles at the locations indicated on Fig. 1. Nitrate only contributes 10% to the total sampled





free-tropospheric aerosol mass, and inorganic nitrate even less so, but an examination of IN's thermodynamic partitioning, a

reversible gas->particle phase transition favored at higher altitudes because of the lower temperatures/higher relative humidities (Nenes et al., 1997), can also illuminate if some of the OA mass loss may also be thermodynamically reversible.

The 24 September, 2016 profile at 12.3°S, 11°E (Fig. 11a; southernmost profile in Fig. 3, top row) is broadly comprised of one main aerosol layer centered on 5 km aged ∼4 days since emission, and a slightly older smoke layer of ∼5 days in age, centered on 3 km (Fig. 11b). The younger aerosol aloft is consistent with the stronger upper-altitude winds. These also transport

moisture, consistent with climatological expectations (Adebiyi et al., 2015; Pistone et al., 2021), generating relative humidities exceeding 80% above 4 km when combined with the cooler high-altitude temperatures (Fig. 11a). The water vapor mixing ratio and statically-stable potential temperature profiles indicate little mixing of air between different altitudes (Fig. 12a). Both the $NO_3$:BC and OA:BC mass ratios increase with altitude (Fig. 11b). The inorganic nitrate fraction is approximately 20%, also increasing slightly with altitude. This profile, also shown in Redemann et al. (2021), spawned the initial speculation that

a large-scale recirculation and subsidence pattern could explain the reduced OA:BC mass ratio at lower altitudes, although the age difference of only one day cannot explain subsidence of more than approximately 500 m.

The 8/31/2016 profile, occurring further south (16.4°S, 6.5°E), sampled aerosol that was constrained near the coast (Fig. 2, top row) by an impinging mid-latitude disturbance (Ryoo et al., 2021), a meteorological condition that is common in September when the AEJ-S is less strong (Zhang and Zuidema, 2021). The associated synoptically-driven ascent mixes moisture upward,

generating a linear increase in water vapor mixing ratio (Fig. 12b). The relative humidity increases with height (Fig. 11c), reaching 80% and capable of generating mid-level clouds elsewhere (Adebiyi et al., 2020). The aerosol aloft is older, at approximately 9 days in age above 3.5 km, overlying younger aerosol aged between 5-6 days below 3 km (Fig. 11d). Winds are weak below 4 km, and primarily eastward throughout the full profile (Fig. 11c). For this profile, the OA:BC and $NO_3$:BC mass ratios also increase with altitude, but not as strongly as on 9/24. The inorganic nitrate fraction is slightly higher in the

mean than on 9/24 (∼25% versus ∼20%). This difference between the two days is consistent with a loss of organic nitrate with age, although the smoke emissions at the source could have also contained more inorganic nitrate initially on 8/31/2016 than on 9/24/2016.

For the 9/24 profile, the increase in the inorganic nitrate fraction with height of approximately 0.05 can only be because the lower temperatures and higher relative humidities above 4 km favor the particle phase (Nenes et al., 1997; Zhang et al., 1999),

as the increase is inconsistent with younger aerosol aloft being organic-nitrate-enriched. Although the older aerosol aloft on 8/31 should support an even more pronounced vertical structure in IN:$NO_3$ than on 9/24, this is not the case. More consistent with the hypothesis that continuing aerosol aging favors a net aerosol loss, is that the overall increase in the $NO_3$:BC and OA:BC mass ratios with height, which could contain some thermodynamical repartitioning, is less pronounced on 8/31 than on 9/24. It may be that the signal-to-noise ratio is too small to resolve the IN vertical structure well within these individual

profiles. This analysis does clarify that most of the nitrate contained within the sampled BBA is organic in nature, in contrast to more aged aerosol sampled at Ascension (Wu et al., 2020).

In these two examples, the younger aerosol occupies a more humid environment on 9/24, and a drier environment on 8/31. A compositing of OA:BC, $NO_3$:BC and aerosol age by RH for all six flights reveals younger aerosol is more likely to occupy





a more humid environment than does older aerosol (Fig. 13). The mean $NO_3$:BC ratio decreases by almost 50% as the RH

decreases from  70% to  30% (Fig. 13a), consistent with a thermodynamic repartitioning. For the same data samples, the mean

OA:BC mass ratio decreases from $10.5 \pm 2.6$ for RH values between 60-80% to $9.9 \pm 2.1$ for 20% < RH < 40%. This indicates

that a thermodynamical repartition can only explain a relative decrease in OA:BC with age of less than 10%. We conclude that

most OA loss in the southeast Atlantic free troposphere is irreversibly lost after 4 days, with thermodynamic repartitioning

providing a minor contribution (dilution and wet removal processes are already excluded by construction). This finding is not

new in qualitative terms, with prior field campaigns in remote areas also highlighting a net OA loss as BBA ages beyond a day

(e.g., Capes et al., 2008; Jolleys et al., 2012, 2015; Hodzic et al., 2015; Konovalov et al., 2019). Causes may be an evaporation

of the organic material (Jolleys et al., 2015) as part of a continuing oxidation. The 9/24 example indicates how the advection

of relatively fresher aerosol at higher altitudes can accentuate the observed vertical structure in nitrate and organic aerosol to

black carbon, through a combination of both the thermodynamical repartitioning and continuing loss of organic nitrate and

aerosol, whereas the vertical structure in OA:BC and $NO_3$:BC in the 8/31 profile is less pronounced, which can be interpreted

as a compensation between the thermodynamical repartitioning and chemical loss of OA over time.

## 7    Radiative implications and inferences

The low OA:BC ratio of southern African BBA emissions in September, combined with a subsequent loss of OA as the

smoke plume advects westward, has implications for the single scattering albedo (SSA, the ratio of the scattering efficiency

to extinction efficiency). Fig. 14 indicates the SSA values at 530 nm range from 0.83 to 0.89, and that they vary strongly

with the OA:BC ratio. SSA can be parameterized as a function of OA:BC as $SSA_{530nm}$=0.801+0.0055*(OA:BC) (Fig. 14),

with a correlation coefficient of 0.7. The SSA values are consistent with Pistone et al. (2019), who report ORACLES-2016

mean SSA values at 530 nm of 0.86 based on all the flight data, at all altitudes, with an inter-quartile range of approximately

0.028. As shown within Pistone et al. (2019), the SSA values reported here are lower than those documented over land or

closer to the coast during SAFARI (Haywood et al., 2003; Formenti et al., 2003), and slightly less than Dubovik et al. (2002)

based on AERONET-derived column-average measurements of BBA over continental Africa. They are on par with AERONET

September-mean values at Mognu (Eck et al., 2013). SSA values reported at Ascension Island, further offshore (Zuidema et al.,

2018; Wu et al., 2020), are lower. The aerosol outflow on 24 September 2016 in the middle free troposphere registered both

the highest OA:BC mass ratio (14.5) and the highest SSA (0.89) of the ORACLES-2016 campaign. An SSA parameterization

on OA:BC provides a straightforward connection between the BBA chemical properties to the modeling of the direct aerosol

radiative effect for this region, with the caveat that incorporation of the remaining ORACLES data would further enhance the

robustness of this relationship. The relationship is not anticipated to hold for the more polluted northern hemisphere, for which

SOA production is expected to be more significant (Jolleys et al., 2012). In addition, such a parameterization is only effective

if the model OA:BC mass ratios are realistic. GIven that OA:BC mass ratios are often too low in models, their absorption of

sunlight will also be overestimated (Brown et al., 2021) until the chemical composition is correctly modeled.



Another SSA parameterization is put forth within Brown et al. (2021) based on the BC to total carbon (TC) mass ratio, namely $SSA_{550nm}$=0.969-0.779*(BC:TC), where TC=BC+organic carbon (OC) and OC is estimated from OA:OC=1.26*O:C+1.18 (Aiken et al., 2008). We find a weaker dependence on BC:TC: $SSA_{530nm}$=0.929-0.389*(BC:TC) (Fig. 14b; $r$=-0.79) using the same calculation for OC. The weaker relationship reflects this study's smaller range of BC:TC values (0.12-0.22 versus 0-0.3

for Brown et al. (2021)) and lack of highly-scattering aerosol.

Of further interest is whether any brown carbon absorption of sunlight is occurring, meaning a wavelength-dependent contribution to the total solar absorption that is typically small but potentially significant. Brown carbon is associated with primary organic aerosol, as SOA is typically considered to only scatter sunlight (Laskin et al., 2015). By 5 days, the primary organic aerosol that may contribute to wavelength-dependent solar absorption is expected to be gone, with SOA mostly scattering light

(Bond and Bergstrom, 2006). Nevertheless, some studies suggest oxidation can result in new chromophrores (O'Brien and Kroll, 2019), with Carter et al. (2021) suggesting that brown carbon absorption may be relatively higher for the southeast Atlantic as a function of OA:BC than for the more OA-enriched fire emissions of the western United States, based on laboratory findings that BrC absorption increases with BC:OA mass ratios (or decrease with OA:BC) (Saleh et al., 2014; McClure et al., 2020).

Indeed, Ozone Monitoring Instrument UltraViolet Aerosol Index values for August-September 2017 suggest brown carbon should be present east of the prime meridian (Carter et al., 2021). The AAE values (470-660nm), calculated from the level legs occurring within the individual flights, reach approximately 1.2 closer to the coast and in several locations further west (Fig. 15). AAE increases weakly with OA:BC (correlation coefficient $r$ of 0.27). Although not conclusive (little absorption is expected by brown carbon at 470 nm), the correlation and slightly elevated values closer to the coast are consistent with some brown

carbon absorption close to the coast. To be consistent with Bond and Bergstrom (2006) would require some chromophores to be present in the African aerosol outflow after 4 days since emission, perhaps as a byproduct of oxidation (O'Brien and Kroll, 2019). The model-derived age estimates indicate most of the aerosol was emitted at least six days previously, with that on 9/24/2016 being the youngest at 4-5 days corresponding to the lowest $f44$ values. At this time since emission, all primary organic aerosol should be consumed, as well as brown carbon (Laskin et al., 2015).

Another assessment of brown carbon absorption can be done using the mass absorption coefficients at 470 nm relative to the sum of the BC and OA mass concentration (Carter et al., 2021), shown for the same level legs in Fig. 16. These range from 0.94 closer to the coast to 1.73 m² g⁻¹ further to the northwest, decreasing with increasing OA:BC ($r$=0.86) as expected. These values are less than the median value of approximately 1.75 m² g⁻¹ reported in Carter et al. (2021) for the 2016 ORACLES deployment. The $MAC_{BC+OA}$ values closest to the coast, range between 0.94 to 1.2 m² g⁻¹. This is significantly

less than the median value of 1.51 m² g⁻¹ reported in (Carter et al., 2021) based on the Saleh et al. (2014) parameterization, suggesting that the Saleh et al. (2014) parameterization may be overestimating brown carbon absorption over the southeast Atlantic. The discrepancy could be explained by different data treatments, with this study only selecting data from level legs with relatively homogeneous aerosol characteristics, so as to increase the signal-to-noise ratio and not convolve results with differences attributable to vertical structure.





The level-leg mean mass absorption coefficient at 660 nm relative to the black carbon mass concentration alone is 10.8 m$^2$ g$^{-1}$, slightly higher than the median value of 9.3 m$^2$ g$^{-1}$ reported for ORACLES-2016 by Carter et al. (2021) and less than the CLARIFY median value of 11.5 m$^2$ g$^{-1}$. This is likely because the BC-enriched 31 August 2017 flight contributes strongly to the mean value we report here. The 31 August 2017 flight occurred during the CLARIFY time frame, suggesting that the CLARIFY MAC$_{BC,660nm}$ may also have been elevated because the OA:BC mass ratio of the source emissions was

lower compared to ORACLES values, as opposed to more continued aging. More analysis will be required to come to a more definitive conclusion. More significant is that all of these MAC values exceed the Bond and Bergstrom (2006) value of 6.25 m$^2$ g$^{-1}$ reported for strongly light-absorbing carbon.

    As has been previously noted (Taylor et al., 2020), lensing, by which absorption increases through a Mie effect generated by the OA coating, is in theory able to both decrease the SSA and maintain a constant AAE (Lack et al., 2012; Cappa et al.,

2012). Mie calculations, which require spherical shapes, may overestimate lensing effects, but aged BBA do compact from the fractals that can define soot upon emission (Taylor et al., 2020). There is some indication that the co-emitted sulfate can contribute to the lensing (Christian et al., 2003), as can the enhanced humidity present within the aerosol layers (Redemann et al., 2001). Combined with other absorptive coating characteristics (Denjean et al., 2020), and photo-bleaching (Taylor et al., 2020), this may explain why the BBA over the remote southeast Atlantic is more light-absorbing than noted elsewhere without

requiring the presence of brown carbon. We have not examined the impact of particle size and this may also contribute to the explanation (smaller particles scatter less light according to a size$^4$ dependence).

## 8   Conclusions

In this study we attempted to place on firmer footing an early interpretation made during the ORACLES 24 September 2016 research flight, in which an increase in the single-scattering albedo with height was associated with a smoke plume enriched in

organic aerosol mass relative to black carbon mass (Redemann et al., 2021). The OA-enriched aerosol layer was sampled close to the coast at an altitude of 4-6 km within strong westward winds, and overlaid a lower, older aerosol layer of a lower OA:BC mass ratio. In the field, the hypothesis was made that loss of OA through chemical aging while the aerosol was recirculated anticyclonically to the African continent, subsiding slowly in the meanwhile, could explain the pronounced vertical structure in aerosol composition and thereby in the single-scattering albedo (Redemann et al., 2021). If a net OA loss is characteristic of

BBA over the southeast Atlantic once the aerosol has advected away from the continent, this contributes to the explanation for why the BBA over the SEA is so absorbing of sunlight in September.

    When controlled for variability in the source emissions through the use of the BC:CO mass ratio, a decrease in OA:BC of 25-35% could be attributed to chemical aging. A simple SSA parameterization based on OA:BC of SSA=0.801+0.0055*(OA:BC), indicates that the range of OA:BC of 8 through 14 equates to an SSA variability of 0.83 to 0.89. A 25% change in OA:BC

through chemical aging corresponds to a relative SSA change near 0.1, or almost 20% of the observed variability. We attribute the remaining 80% to variations occurring in the production of SOA near the emission source, based on for example an increased availability of precursor gases when BC:CO ratios are lower (Yokelson et al., 2009; Vakkari et al., 2018).





The aerosol sampled during ORACLES-2016 and 31 August 2017 possessed modified combustion efficiencies exceeding 0.975, some of the highest reported in the literature surveyed for Table 1, with the exception of Ascension Island (Wu et al., 2020). Such values are characteristic of savanna grasses (Janhäll et al., 2010; Vakkari et al., 2018) that are also dry (Korontzi et al., 2003), which are known to possess higher BC efficiencies (Andreae, 2019). For this region far removed from urban and industrial sources of pollution, continued production of SOA after 1-2 days is expected to remain minor (e.g., O'Brien and Kroll, 2019), with other work also reporting a small net OA loss with time for smoke from northern Africa (Jolleys et al., 2012, 2015). Recent analysis of filters from August 2017 during ORACLES and CLARIFY also consistent with continued OA loss (Dang et al., 2021). This contrasts with northern hemisphere fire emission sources. Brown carbon production has been linked to low OA:BC ratios (Saleh et al., 2014; McClure et al., 2020). We do not see much evidence for brown carbon in the ORACLES-2016 AAE and MAC values, perhaps because brown carbon is more closely linked to primary than to secondary organic aerosol, although we are limited by optical measurements that do not extend to wavelengths smaller than 470 nm.

At face value, this works further supports the use of optical parameterizations upon chemical parameters for improving the radiative representations of BBA in global aerosol models with increasingly sophisticated SOA schemes (e.g., Lou et al., 2020). Such parameterizations place more pressure on producing realistic representations of the BBA composition, however. This study adds to literature indicating that OA model estimates made by multiplying the organic carbon by a factor of 1.4 will underestimate OA in this (and other) regions (Aiken et al., 2008; Tsigaridis et al., 2014; Shinozuka et al., 2020; Doherty et al., 2021), so that SSA parameterizations based on OA:BC or related chemical parameters will overestimate solar absorption (Brown et al., 2021). This study's OA:OC mass ratios of $2.2 \pm 0.1$ has also been shown for the Atomic Tomography mission (Hodzic et al., 2020). Modeled OA:BC mass ratios can also be overestimated by over a factor of two over the southeast Atlantic in global models with sophisticated aerosol schemes (Chylek et al., 2019), suggesting the loss of OA with aging or slower SOA production processes (Kroll and Seinfeld, 2008; McFiggans et al., 2019) can also be under-accounted for. This is more important for remote environments containing thick smoke layers lacking the precursor gases for additional SOA production. Southern Africa produces approximately one-third of the world's carbon emissions through biomass-burning (van der Werf et al., 2010), with the global majority of the absorbing aerosols above cloud occurring above the southeast Atlantic (Waquet et al., 2013), indicating the importance of realistic representations for this radiative climate (Mallet et al., 2021).

September is a unique transition month. This is when the AEJ-S is strong, because the thermal gradient between the Kalahari heat low and the cooler, moist equatorial climate is more pronounced, driving the upper-level winds that transport the aerosol (Adebiyi and Zuidema, 2016; Kuete et al., 2020). The winds occur to the north of the heat low, with only dry convection lofting the aerosols generated by the burning of the flaming-efficient dry grasses. The winds distribute aerosol with low OA:BC mass ratios as far away as south America (Holanda et al., 2020), so that the entire south Atlantic is covered by a blanket of highly-absorptive aerosol. This has a pronounced climate impact, because the south Atlantic stratocumulus deck is also at a peak extent and thickness then (Zuidema et al., 2016a), reinforced in part by the radiative stabilization of the south Atlantic free troposphere. Although IPPC AR5 assessments suggest the ability of smoke to both scatter and absorb sunlight leads to a net compensation globally, this is not the case for the southeast Atlantic.



*Data availability.* The data are available through doi=10.5067/Suborbital/ORACLES/P3/2016_V2 and doi=10.5067/Suborbital/ORACLES/P3/2017_V2

*Author contributions.* The present work was conceived by P.Z., P.S, S.H. and A.D. S.F. contributed to the HiGEAR data analysis, A.S.
provided the BC datasets and P.S. the WRF-AAM model age estimates. Portions of this work first appeared in the M.S. thesis of A.D at U. of Hawaii. All authors contributed to the final writing.

*Competing interests.* The authors declare no competing interests.

*Acknowledgements.* ORACLES is a NASA Earth Venture Suborbital-2 investigation, funded by the US National Aeronautics and Space
Administration (NASA)'s Earth Sciences Division and managed through the Earth System Science Pathfinder Program Office (grant no.
NNH13ZDA001N-EVS2). This work was further supported by the US Department of Energy (DOE: grant DE-SC0018272 to P.Z. and P.S. and DE-SC0021250 to P.Z. We thank Hugh Coe, Huihiu Wu and Jonathan Taylor for initial conversations that helped guide this work.



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





**Table 1.**

| Fuel/Geographic Source | BC:CO*$10^{-3}$ | reference |
|---|---|---|
| savannah | 2-15 | Vakkari et al. (2018) |
| grass | 10-17 | Vakkari et al. (2018) |
| savannah | 7.9 | Andreae (2019) |
| agriculture | 5.6 | Andreae (2019) |
| savannah | 5.9 | Akagi et al. (2011) |
| agriculture (crop residue) | 7.4 | Akagi et al. (2011) |
| NW African agriculture, smouldering | 7.2 | Capes et al. (2008) |
| southern Africa (SAFARI) | 7.0 | Formenti et al. (2003) |
| Ascension Island, August | 8.7-11.4 | Wu et al. (2020) |
| **this study** | 9.6 | |

all BC:CO values are dimensionless. Most CMIP6 models rely on the Akagi et al. (2011) emission factors.



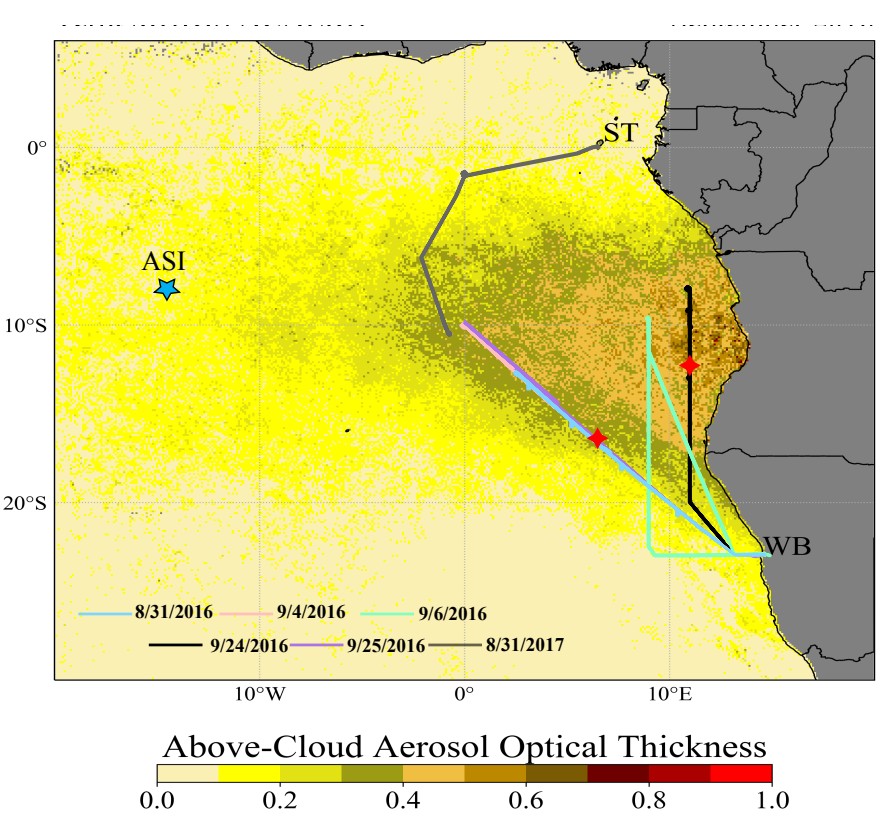

**Figure 1.** Terra MODIS Above Cloud Aerosol Optical Depth (Meyer, 2015) for September 2016 overlaid with the tracks of the 6 flights selected for this study. Locations of the profiles shown in Figs. 10-11 are indicated with red diamonds. ST=Sao Tome; WB=Walvis Bay; ASI=Ascension Island.

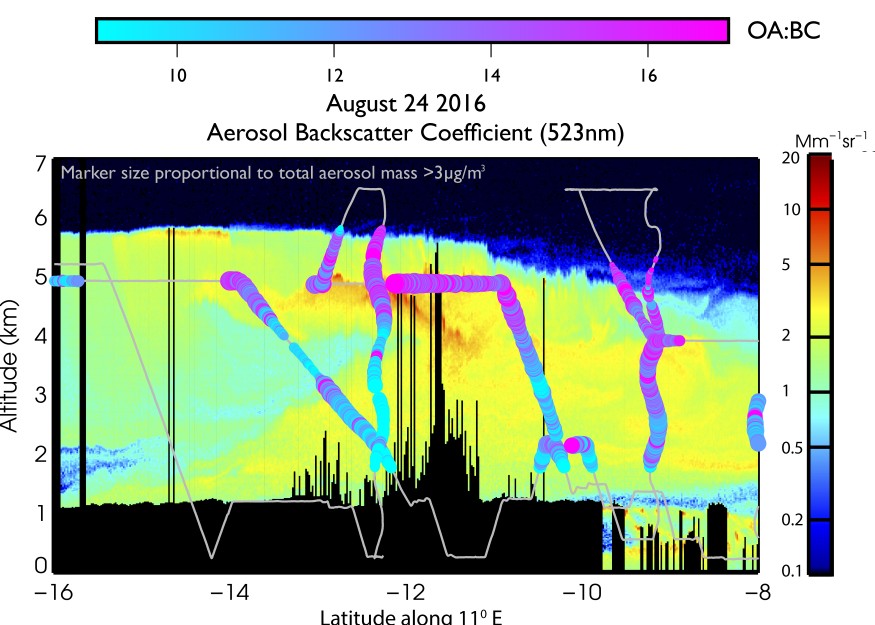

**Figure 2.** September 24 2016 flight track with colorized OA:BC mass ratios superimposed on High Spectral Resolution Lidar-2 523 nm aerosol backscatter imagery collected along 11°E, near in time to the P-3 plane location's at 10°S from an overflying ER-2 plane.

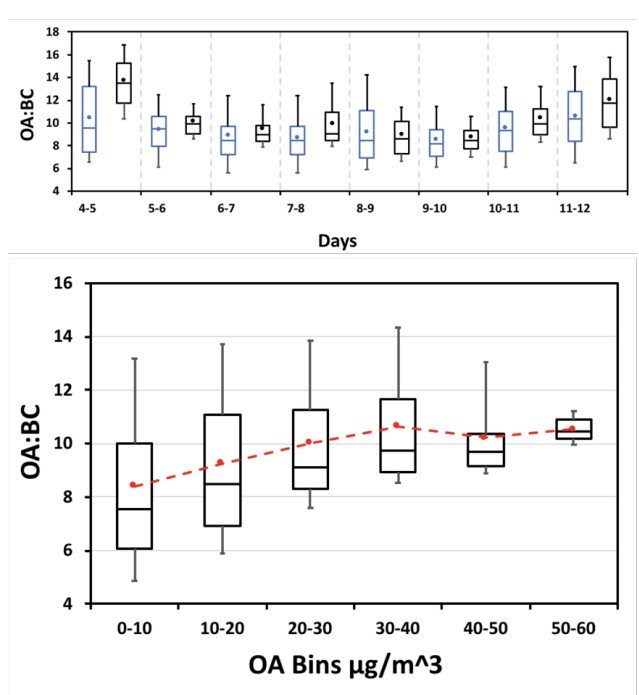

**Figure 3.** a) OA:BC as a function of model age for OA > 3 $\mu$g m$^{-3}$ (blue) and OA >20 $\mu$g m$^{-3}$ (black). b) OA:BC composited by aerosol mass bins, shown using 10, 25, 50, 75 and 90th percentiles, with means in red.



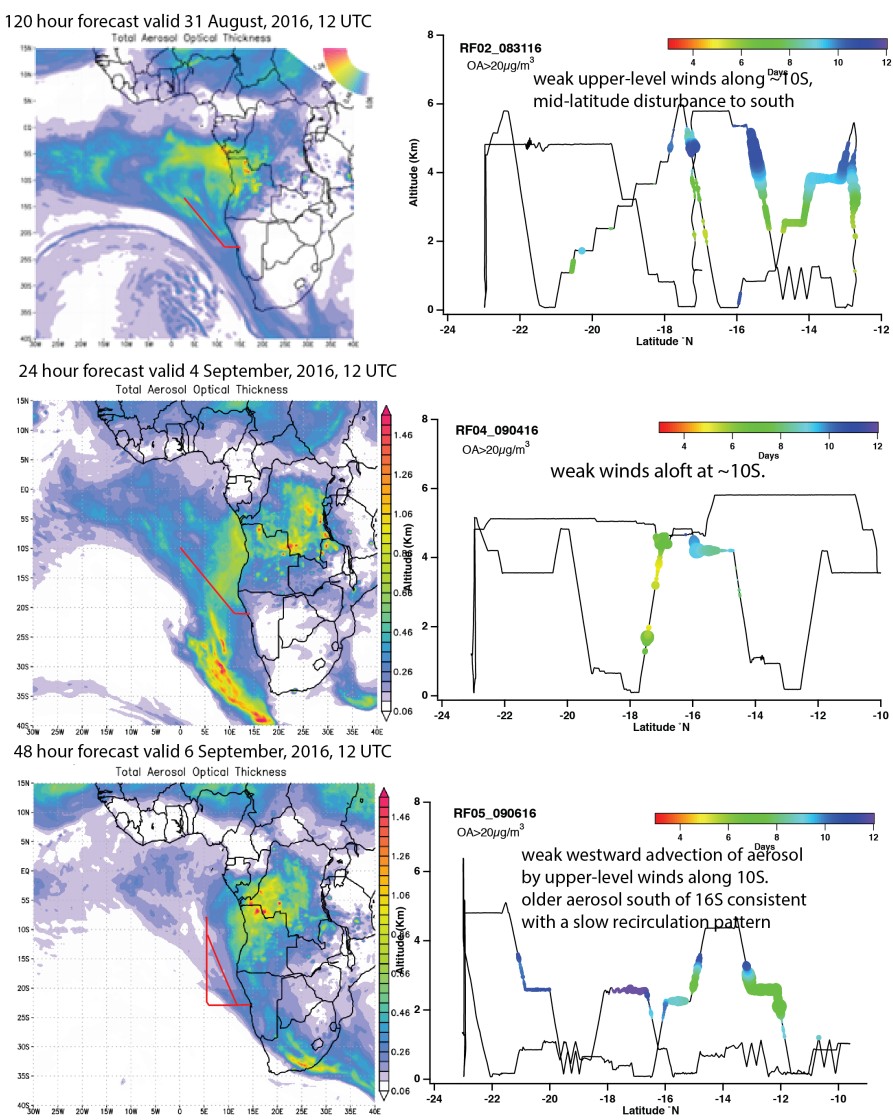

**Figure 4.** left) GMAO (spell) aerosol optical thickness forecasts for 31 August, 2016; 4 September, 2016; 6 September, 2016. Right) Altitude versus latitude cross-sections of the flights overlain with the colorized aerosol age, with the size of the marker providing a qualitative marker of aerosol mass.



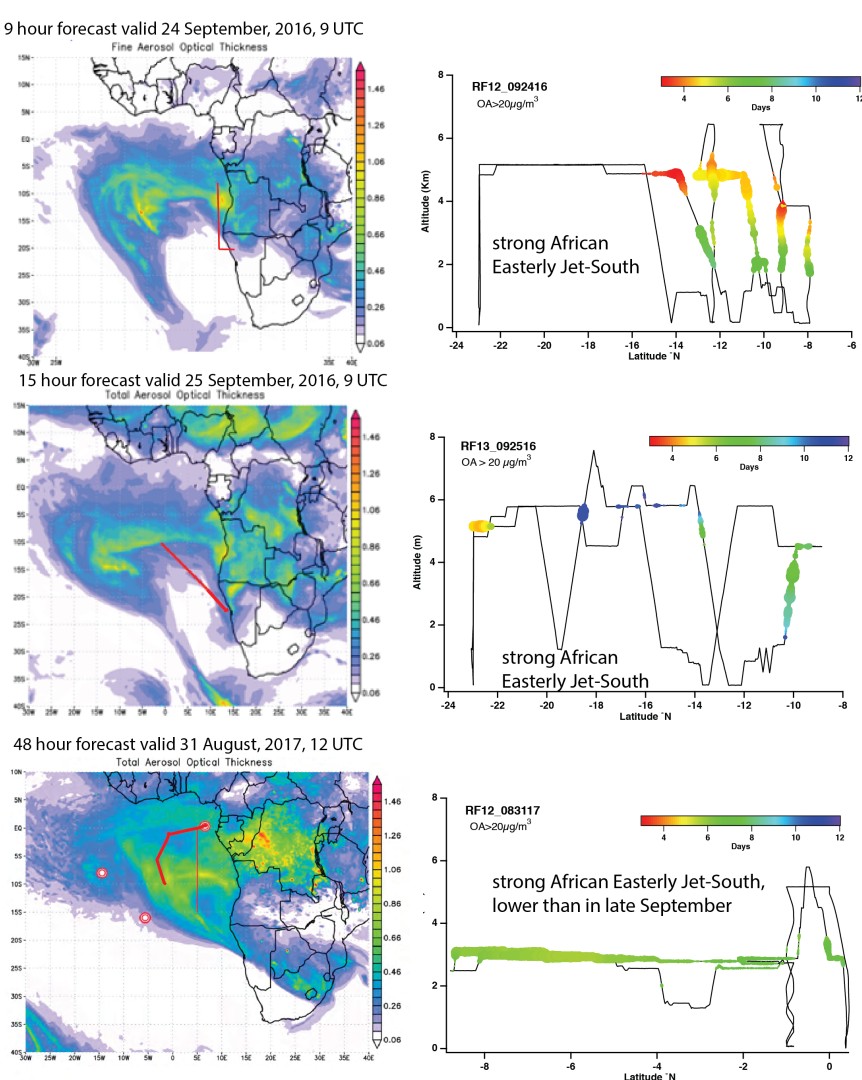

**Figure 5.** left) same as Fig. 3 but for 24 September, 2016; 25 September, 2016, and 31 August, 2017.



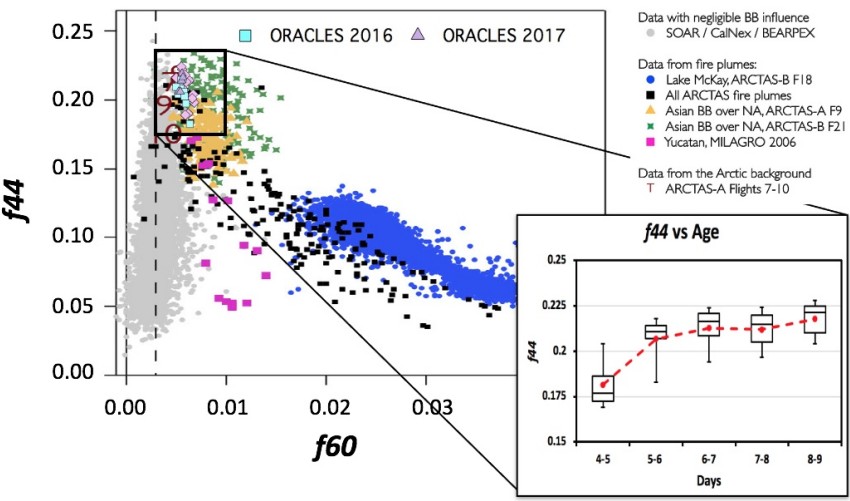

**Figure 6.** left) *f44* versus *f60* for multiple campaigns, adapted from (Fig. 4 Cubison et al., 2011), overlaid with flight-averaged values from ORACLES (August 2017-grey filled triangles; 2016-light blue filled squares; these include more than just the 6 flights selected for this study). Numbered flights from ARCTAS-A represent 2-week old Siberian biomass burning smoke sampled in northern America, examined in more detail within Cubison et al. (2011). right) ORACLES *f44* values as a function of model-derived age, shown using 10th, 25th, median, 75th and 90th percentiles (box-whisker plots) and mean values (connected red filled circles), for the 6 selected flights, for OA>20 $\mu$g m$^{-3}$.



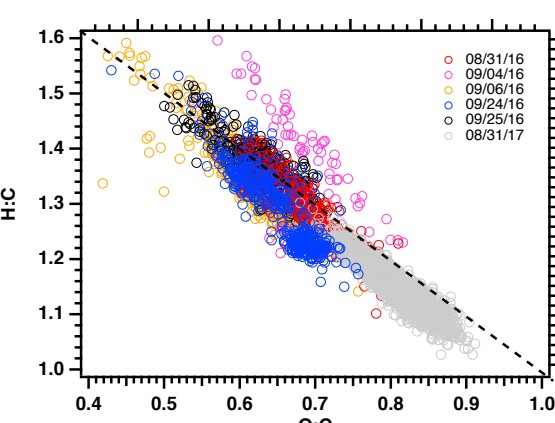

**Figure 7.** Hydrogen to carbon (H:C) mass ratio versus oxygen to carbon (O:C) mass ratio, colorized by flight, shown at the native 5-second time resolution.

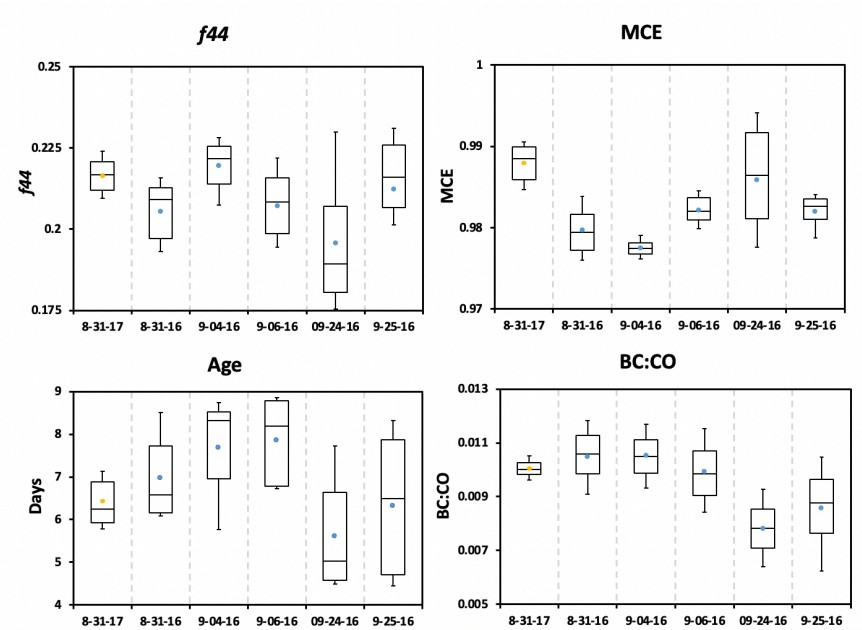

**Figure 8.** a) *f44*, b) Modified combustion efficiency (MCE), c) model-derived time since emission (age), and d) black carbon to carbon monoxide (BC:CO) ratios for indicated flights. Whiskers represent the 10th and 90th percentiles, boxes illustrate the 75th and 25th percentiles with a line indicating the median and yellow (2017) and blue (2016) filled circles representing the mean. OA>20 $\mu$g m$^{-3}$ only.



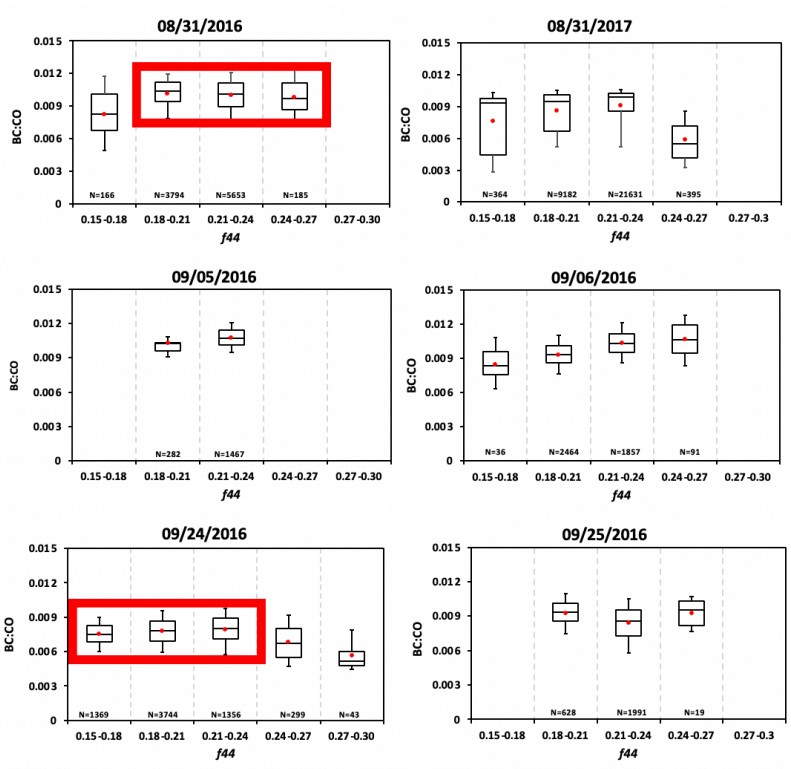

**Figure 9.** BC:CO ratios (dimensionless) as a function of *f44* for the six flights. Whiskers represent the 10th and 90th percentiles, boxes illustrate the 75th and 25th percentiles with a line indicating the median and a red filled circle the mean. OA>20 $\mu$g m$^{-3}$ only. The number of 1-second samples contributing to each *f44* bin of each flight is also indicated. The data selected for further analysis within two case studies are outlined in red.

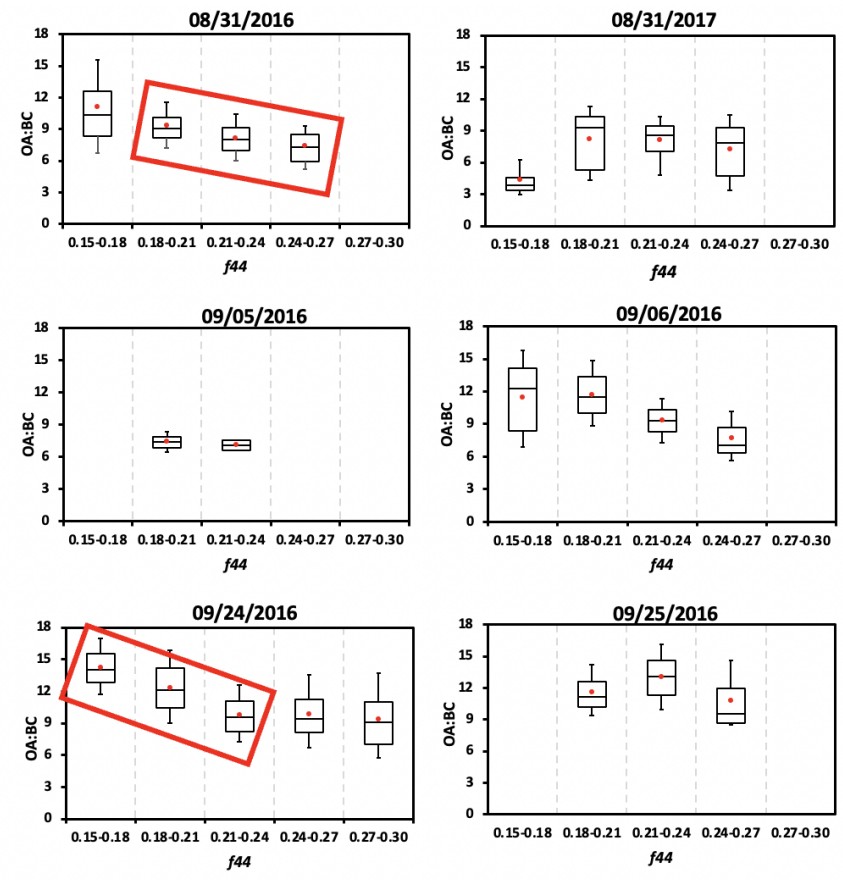

**Figure 10.** OA:BC mass ratios as a function of *f44* for the six flights. Whiskers represent the 10th and 90th percentiles, boxes illustrate the 75th and 25th percentiles with a line indicating the median and a red filled circle the mean. OA>20 $\mu$g m$^{-3}$ only. The data outlined in red correspond to the same samples outlined in red in Fig. 9.



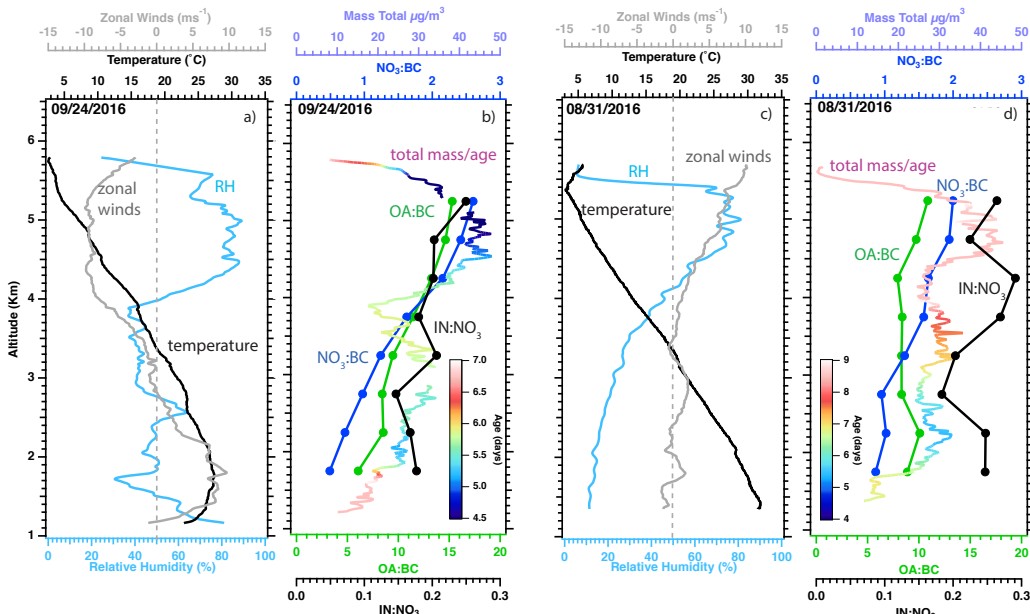

**Figure 11.** 24 September, 2016 (12.34°S, 11°E) vertical profiles of a) relative humidity (%; blue), zonal winds (m s$^{-1}$; grey) and temperature (°C), and b) the inorganic nitrate to black carbon mass ratio (IN:BC; black), organic aerosol to black carbon mass ratio (OA:BC; green), total nitrate to black carbon ratio (NO$_3$:BC; blue) averaged every 500 m (approximately 2 minutes of data), and total mass concentration (OA + BC + SO$_4$ + NO$_3$ + NH$_4$ in $\mu$g m$^{-3}$; 1Hz resolution) colored by aerosol age c)-d): same as a)-b) but for 31 August, 2016 (16.4°S, 6.5°E). Note the different color scales for aerosol age.

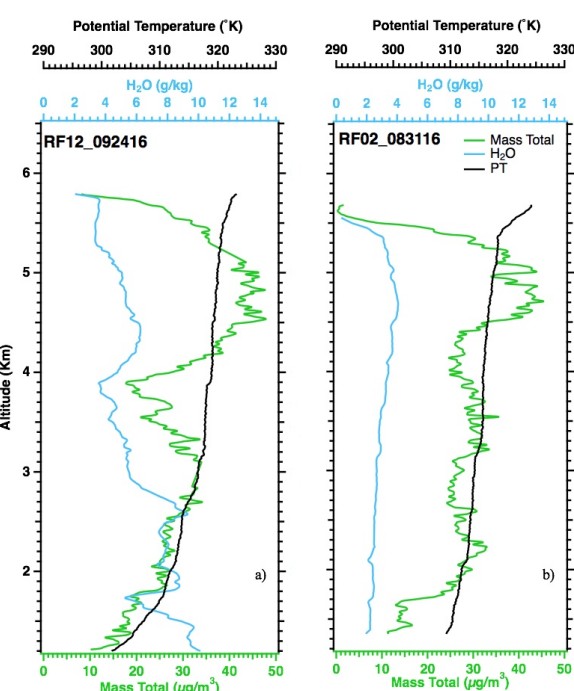

**Figure 12.** Water vapor (blue), potential temperature (black), mass total concentration (OA+SO$_4$+NO$_3$+NH$_4$+BC) (green) vertical profiles for left) 24 September, 2016 (12.34°S, 11°E) and right) 31 August, 2016 (16.4°S, 6.5°E).



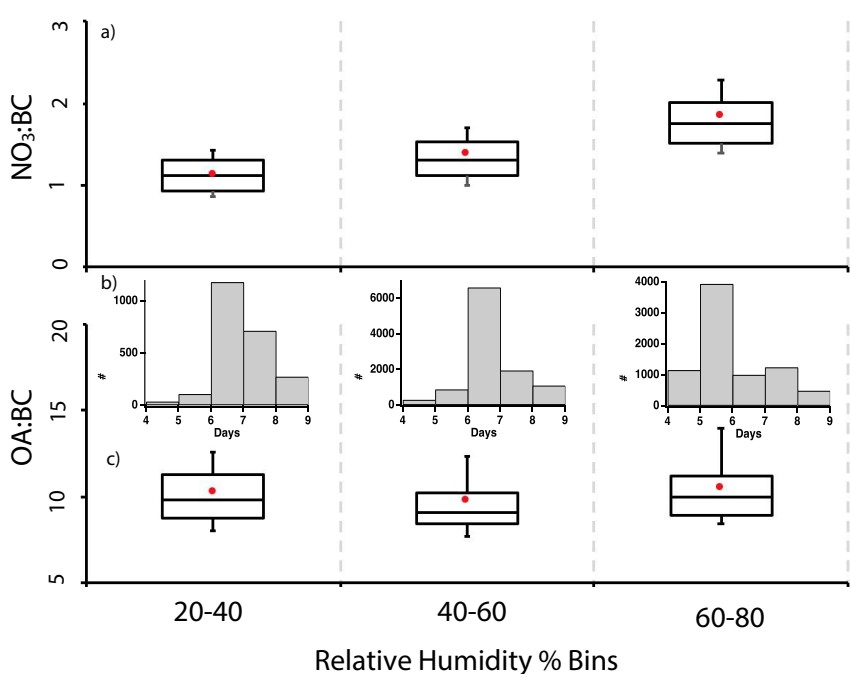

**Figure 13.** a) NO$_3$:BC and c) OA:BC mass ratios for the 6 selected flights as a function of relative humidity, for OA>20 $\mu$g m$^{-3}$ at STP. The 10th, 25th, median, 75th and 90th percentiles are indicated using box-whiskers, the mean with solid red circle and marker. b) corresponding distribution of aerosol ages within each relative humidity range, with the y-axis indicating the number of 1-sec samples.



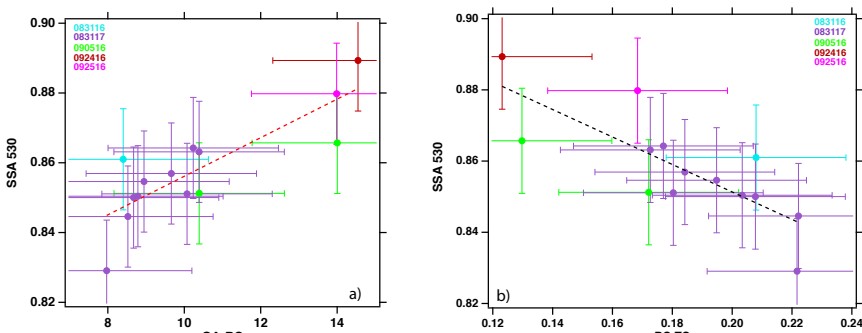

**Figure 14.** a) Level-leg-mean ± standard deviation values for SSA$_{530nm}$ versus the OA:BC mass ratio, colored by flight. The best-fit line is represented by SSA=0.801+0.0055*(OA:BC). b) same as a) but for SSA$_{530nm}$ versus the BC:TC mass ratio, where TC=BC+organic carbon. The best-fit line is SSA= 0.93-0.39*(BC:TC), with a correlation coefficient of -0.79. Times and spatial ranges of the level-legs provided in Table S1.

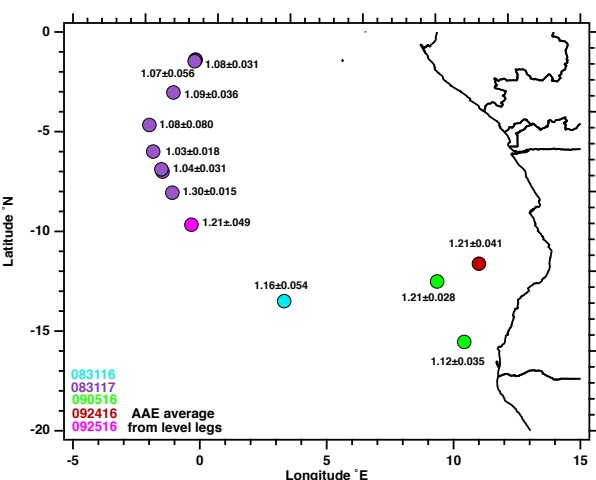

**Figure 15.** Absorption Angstrom exponent (470-660 nm) for the same level legs shown in Fig. 14, similarly colorized by flight.

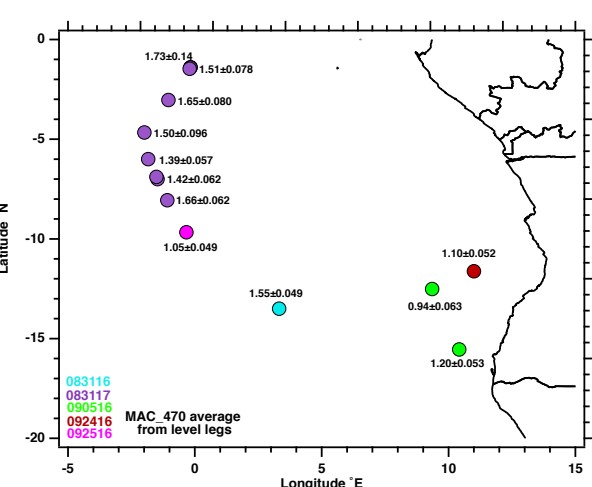

**Figure 16.** Mass absorption coefficient (MAC$_{OA+BC}$, in units of Mm$^{-1}$/($\mu$g m$^{-3}$) at 470nm for the same level legs shown in Fig. 14, similarly colorized by flight.