# Peer review of "Non-reversible aging can increase solar absorption in African biomass burning aerosol plumes of intermediate age"

_Atmospheric Chemistry and Physics, 2021_

## Referee Comment (RC1)

Review of acp-2021-1081

Dobracki et al., Non-reversible aging can increase solar absorption in African biomass burning aerosol plumes of intermediate age

The authors discuss results related to an aircraft campaign that sampled biomass burning aerosols off the coast of western Africa. They present a nice description of the relevant instrumentation and their approach to data analysis and interpretation. However, I do have some comments related to the approach and subsequent interpretation that I feel are critical for the authors to address prior to acceptance.

General Comments

The authors restricted their analysis to data with OA > 20 $\mu$g m$^{-3}$. This leads to several questions:

1. If the OA:BC values in Figure 3a are "significantly less" for OA > 3 $\mu$g m$^{-3}$ compared to OA > 20 $\mu$g m$^{-3}$, what are the *p*-values for these comparisons?
2. What criterion is being used to establish "stabilization" of OA:BC? When I look at Figure 3b, I see median values that continue to gradually increase, similar to, e.g., Figure 6 in May et al. (2015). Hence, this threshold appears to be subjective, and therefore, at best, weakly justified.
3. Why does this threshold matter? Based on the text in Lines 126 – 137, it seems like uncertainty? Even ignoring the ~35% uncertainty in the AMS and ~20% uncertainty in the SP2, e.g., Laborde et al. (2012) – which I don't see a discussion of in this work – there is a lot of variability in the data, cf. the authors' Figure 3. This is likely some combination of inter-fire variability and intra-flight variability.
4. What, if any, difference would a relaxation of this threshold have on the conclusions of the study?

Synthesizing some of the information presented, it is unclear how the conclusion presented by the title of the manuscript can be reached. The logic appears to follow: OA is removed from the aerosols, so the SSA decreases; therefore, the aerosols absorb more light. Ignoring any brown carbon or lensing influences on absorption, the magnitude of light absorbed would be the same if the BC concentration does not change! Conversely, I would agree with logic that follows: OA is removed from the aerosols, so the SSA decreases; therefore, the aerosols scatter less light.

There are some sections in the text where it feels like it's a bit dense. For example, the coupling of meteorological conditions to the airborne aerosol observations is nice, but I am wondering if the readability could be improved by providing a high-level discussion in the main text and moving some of the details to the supplemental material. (This is perhaps the least essential of my comments, so if the authors wish to ignore it, so be it)

Specific Comments

Lines 10 – 11: This suggests that only two flights had constant BC:CO, while Lines 83 – 84 indicate that both BC and CO are conserved after emission. Is this only true for two flights due to some mixing effect during transport?

Lines 28 – 29: Are the authors considering CO to be a greenhouse gas because it influences atmospheric chemistry? To my knowledge, it is a very weak absorber.

Lines 38 – 42: I am curious as to why the authors cite a "general" reference for SOA, when a number of studies have focused on this, specifically for biomass burning – (Bian et al., 2017; Grieshop et al., 2009; Hennigan et al., 2011; Ortega et al., 2013; Vakkari et al., 2014, 2018) to name a few. Conversely, there are a number of specific references that appear to be supporting a point related to net loss of OA from the particles during aging.

Line 50: Through a document search, I was unable to find a definition for BBA (biomass burning aerosols?)

Line 51: Why does the wavelength of light matter? Is this simply because values are often reported somewhere between 530 and 550 nm, so calling all of these "green" avoids this issue? (which is totally fine!)

Lines 116 – 118: My understanding, based on laboratory observations but grounded by combustion chemistry, is that flaming combustion produces more BC, while smoldering produces more OA. Given that OA tends to be emitted in greater quantities than BC in the real world, maybe the statement is okay as written.

Lines 138 and 141: Should these be references to Figure 3?

Line 250: A molecule with a formula of $C_6H_5OH$ is not a carboxylic acid.

Lines 253 – 261: Do the authors have any estimate of, e.g., OH radical concentration or ozone concentration for the ORACLES flights? Presumably a more oxidative environment could also result in a larger OA:OC ratio. This seems to be consistent with what the authors write in Lines 263 – 265.

Lines 299 – 306: I have two comments related to this paragraph.
1. Why would BC:CO change with aging? The authors stated that these are both conservative tracers in Lines 83 – 84, so what could happen so these would not be conserved?
2. I would argue that the evidence could support continued evaporation of OA – less oxidized semi-volatile material could evaporate with continued dilution of the plume, leaving more oxidized (extremely) low-volatility material in the aerosol phase, in which case, this would not be "non-reversible". (although the authors are forcibly excluding dilution-driven evaporation from their analysis by only considering OA > 20 μg m$^{-3}$).

Section 6: There appears to be a substantial focus on nitrate, even though the authors comment that total nitrate is ~10% of the aerosol mass. I would argue that the explanation is more robust if grounded in OA. What I observe in the authors' Figure 11, at least for the 24 Sept. 2016 data is that temperature decreases; therefore, relatively more OA should partition into the particle phase. Figure 1 models this using the reported parameters in May et al. (2013). Moisture does not factor into this parameterization, and the figure below does not capture trends in OA:BC.

[Figure]

*Figure 1. OA gas-particle partitioning as a function of OA concentration at two different temperatures.*

Of course, there may be some "complications" of using a biomass-burning-specific parameterization, because after days of transport, the smoke likely not only undergone chemical aging but also mixing with other aerosol sources. Moreover, an "OA-centric" argument may be less appropriate for 31 Aug. 2016, as the OA:BC is more constant vertically.

Lines 350 – 351: While these values may be statistically different given the large number of samples that are included in Figure 13b, I would argue that the difference is practically irrelevant – the relative difference between the means is only ~6%, while the relative standard deviation for both groups is ~20-25%.

Lines 366 – 367 & 381 – 385: Given the differences between the parameterizations between the authors work and Brown et al. (2021), how can this be resolved? Moreover, what is the utility of such a parameterization? Most Earth-system models contain some aerosol optics module, which is used to compute absorption and scattering, and hence, SSA.

Lines 396 – 398: A value of 1.2 for BC is within the ranges reported by both Lack and Langridge (2013) for laboratory studies and Liu et al. (2018) in a numerical study. Likewise, the classification scheme presented in Cappa et al. (2016) suggests that in the absence of large particles, light absorption is BC-dominated for values less than ~1.5.

Lines 398 -399: On what basis is the statement "little absorption is expected by brown carbon at 470 nm" grounded? I would expect there to be more absorption at that wavelength than at, e.g., 660 nm, given the spectral dependence on light absorption.

Lines 409 – 412: How can the mass absorption cross-section for BC + OA be less than that of BC alone? The authors report a value of ~1 $m^2$ $g^{-1}$ (at 470 nm?), while the value at 470 nm that one would estimate from the Bond and Bergstrom (2006) of 7.5 $m^2$ $g^{-1}$ at 550 nm along with an absorption Ångström exponent of 1 is ~8.8 $m^2$ $g^{-1}$? In light of the reported results in Lines 415 – 422, I'm more confused by this value near 1 $m^2$ $g^{-1}$.

Figure 4: This does not appear to be an "adaptation" of Cubison et al. (2011); it appears as if the authors took Figure 4 from the original, compressed the y-axis, muted the colors, and superimposed their data upon it. Given the relatively narrow window within the $f_{60}$ vs. $f_{44}$ space that these ORACLES data occupy, is this

even necessary? Maybe it's just worth comparing the a subset of the data from Cubison, or even comparing in tabular form.

[Figure]

*Figure 2. Original figure from Cubison et al.*

[Figure]

*Figure 3. The authors' figure.*

References

Bian, Q., Jathar, S. H., Kodros, J. K., Barsanti, K. C., Hatch, L. E., May, A. A., Kreidenweis, S. M. and Pierce, J. R.: Secondary organic aerosol formation in biomass-burning plumes: Theoretical analysis of lab studies and ambient plumes, Atmos. Chem. Phys., 17(8), doi:10.5194/acp-17-5459-2017, 2017.

Bond, T. C. and Bergstrom, R. W.: Light Absorption by Carbonaceous Particles: An Investigative Review, Aerosol Sci. Technol., 40(1), 27–67, doi:10.1080/02786820500421521, 2006.

Brown, H., Liu, X., Pokhrel, R., Murphy, S., Lu, Z., Saleh, R., Mielonen, T., Kokkola, H., Bergman, T., Myhre, G., Skeie, R. B., Watson-Paris, D., Stier, P., Johnson, B., Bellouin, N., Schulz, M., Vakkari, V., Beukes, J. P., van Zyl, P. G., Liu, S. and Chand, D.: Biomass burning aerosols in most climate models are

too absorbing, Nat. Commun., 12(1), 277, doi:10.1038/s41467-020-20482-9, 2021.

Cappa, C. D., Kolesar, K. R., Zhang, X., Atkinson, D. B., Pekour, M. S., Zaveri, R. A., Zelenyuk, A. and Zhang, Q.: Understanding the optical properties of ambient sub- and supermicron particulate matter: results from the CARES 2010 field study in northern California, Atmos. Chem. Phys., 16(10), 6511–6535, doi:10.5194/acp-16-6511-2016, 2016.

Cubison, M. J., Ortega, A. M., Hayes, P. L., Farmer, D. K., Day, D., Lechner, M. J., Brune, W. H., Apel, E., Diskin, G. S., Fisher, J. A., Fuelberg, H. E., Hecobian, A., Knapp, D. J., Mikoviny, T., Riemer, D., Sachse, G. W., Sessions, W., Weber, R. J., Weinheimer, A. J., Wisthaler, A. and Jimenez, J. L.: Effects of aging on organic aerosol from open biomass burning smoke in aircraft and laboratory studies, Atmos. Chem. Phys., 11(23), 12049–12064, doi:10.5194/acp-11-12049-2011, 2011.

Grieshop, A. P., Logue, J. M., Donahue, N. M. and Robinson, A. L.: Laboratory investigation of photochemical oxidation of organic aerosol from wood fires 1: measurement and simulation of organic aerosol evolution, Atmos. Chem. Phys., 9(4), 1263–1277, doi:10.5194/acp-9-1263-2009, 2009.

Hennigan, C. J., Miracolo, M. A., Engelhart, G. J., May, A. A., Presto, A. A., Lee, T., Sullivan, A. P., McMeeking, G. R., Coe, H., Wold, C. E., Hao, W.-M., Gilman, J. B., Kuster, W. C., De Gouw, J., Schichtel, B. A., Collett Jr., J. L., Kreidenweis, S. M. and Robinson, A. L.: Chemical and physical transformations of organic aerosol from the photo-oxidation of open biomass burning emissions in an environmental chamber, Atmos. Chem. Phys., 11(15), doi:10.5194/acp-11-7669-2011, 2011.

Laborde, M., Schnaiter, M., Linke, C., Saathoff, H., Naumann, K.-H., Möhler, O., Berlenz, S., Wagner, U., Taylor, J. W., Liu, D., Flynn, M., Allan, J. D., Coe, H., Heimerl, K., Dahlkötter, F., Weinzierl, B., Wollny, A. G., Zanatta, M., Cozic, J., Laj, P., Hitzenberger, R., Schwarz, J. P. and Gysel, M.: Single Particle Soot Photometer intercomparison at the AIDA chamber, Atmos. Meas. Tech., 5(12), 3077–3097, doi:10.5194/amt-5-3077-2012, 2012.

Lack, D. A. and Langridge, J. M.: On the attribution of black and brown carbon light absorption using the Ångström exponent, Atmos. Chem. Phys., 13(20), 10535–10543, doi:10.5194/acp-13-10535-2013, 2013.

Liu, C., Chung, C. E., Yin, Y. and Schnaiter, M.: The absorption Ångström exponent of black carbon: from numerical aspects, Atmos. Chem. Phys., 18(9), 6259–6273, doi:10.5194/acp-18-6259-2018, 2018.

May, A. A., Lee, T., McMeeking, G. R., Akagi, S., Sullivan, A. P., Urbanski, S., Yokelson, R. J. and Kreidenweis, S. M.: Observations and analysis of organic aerosol evolution in some prescribed fire smoke plumes, Atmos. Chem. Phys., 15(11), 6323–6335, doi:10.5194/acp-15-6323-2015, 2015.

May, A. A., Levin, E. J. T., Hennigan, C. J., Riipinen, I., Lee, T., Collett, J. L., Jimenez, J. L., Kreidenweis, S. M. and Robinson, A. L.: Gas-particle partitioning of primary organic aerosol emissions: 3. Biomass burning, J. Geophys. Res. Atmos., 118(19), 11,327-11,338, doi:10.1002/jgrd.50828, 2013.

Ortega, A. M., Day, D. A., Cubison, M. J., Brune, W. H., Bon, D., de Gouw, J. A. and Jimenez, J. L.: Secondary organic aerosol formation and primary organic aerosol oxidation from biomass-burning smoke in a flow reactor during FLAME-3, Atmos. Chem. Phys., 13(22), 11551–11571, doi:10.5194/acp-13-11551-2013, 2013.

Pokhrel, R. P., Wagner, N. L., Langridge, J. M., Lack, D. A., Jayarathne, T., Stone, E. A., Stockwell, C. E., Yokelson, R. J. and Murphy, S. M.: Parameterization of single-scattering albedo (SSA) and absorption Ångström exponent (AAE) with EC / OC for aerosol emissions from biomass burning, Atmos. Chem. Phys., 16(15), 9549–9561, doi:10.5194/acp-16-9549-2016, 2016.

Vakkari, V., Beukes, J. P., Dal Maso, M., Aurela, M., Josipovic, M. and van Zyl, P. G.: Major secondary aerosol formation in southern African open biomass burning plumes, Nat. Geosci., 11(8), 580–583, doi:10.1038/s41561-018-0170-0, 2018.

Vakkari, V., Kerminen, V.-M., Beukes, J. P., Tiitta, P., van Zyl, P. G., Josipovic, M., Venter, A. D., Jaars, K., Worsnop, D. R., Kulmala, M. and Laakso, L.: Rapid changes in biomass burning aerosols by atmospheric oxidation, Geophys. Res. Lett., 41(7), 2644–2651, doi:10.1002/2014GL059396, 2014.